# Distributed-Satellite-Clusters-Based Spectrum Sensing with Two-Stage Phase Alignment

**DOI:** 10.3390/s22113983

**Published:** 2022-05-24

**Authors:** Yunfeng Wang, Xiaojin Ding, Tao Hong, Gengxin Zhang

**Affiliations:** 1Telecommunication and Networks National Engineering Research Center, Nanjing 210003, China; 2018010212@njupt.edu.cn (Y.W.); hongt@njupt.edu.cn (T.H.); zgx@njupt.edu.cn (G.Z.); 2College of Telecommunications and Information Engineering, Nanjing University of Posts and Telecommunications, Nanjing 210003, China; 3College of Internet of Things, Nanjing University of Posts and Telecommunications, Nanjing 210003, China

## Abstract

We investigate a distributed-satellite-clusters (DSC)-system-based spectrum sensing, to enhance the ability for sensing weak signals. However, the spectrum-sensing performance may be significantly decreased by the phase deviations among different satellite clusters, where the deviations may be caused by the movement and the perturbation of satellites. To eliminate such a decrement, we propose a cooperative spectrum-sensing scheme in the presence of phase deviations, where the deviations are alleviated by a special two-stage phase synchronization. Specifically, the phase compensation is first performed relying on broadcasting reference signals and the ephemeris, to address the challenges of the deviations caused by the movement. Then, a two-bit feedback algorithm, having a dynamic disturbance step size, is further adopted for controlling and mitigating the deviations caused by the perturbation. Additionally, we provide the closed-form expression of the correct detection probability of the proposed spectrum-sensing scheme, using the specially derived probability density function of the sum of the shadowed-Rician random variables with independently identical distribution. Simulation results show that the proposed scheme can achieve the best spectrum-sensing performance, comparing with the traditional energy detection, eigenvalue ratio test and the generalized likelihood ratio test.

## 1. Introduction

Low-orbit (LEO) satellites can perform global spectrum sensing in a seamless way, due to their characteristics of wide coverage and high mobility. However, due to the wide coverage and long transmission distance of a satellite, the signal strength of the sensing target may be weak within a wide beam [1]. Moreover, as a resource-constrained platform, LEO satellites cannot be equipped with large-aperture high-gain antennas, resulting in the poor sensing performance of single satellites. Therefore, cooperative spectrum sensing based on multiple satellites can significantly improve sensing performance by utilizing space diversity and has attracted significant research interest recently [2].

Various cooperative spectrum sensing methods have been proposed, for known noise variance, the “and” “or” criterion based on the popular energy detection (ED), the Roy’s largest-root test (RLRT); for unknown noise variance, the eigenvalue ratio test (ERD), or the generalized likelihood-ratio test (GLRT) [3]. Especially in satellite aided sensing, a trust-weighted cooperative spectrum sensing algorithm based on a satellite cluster was proposed in [4], wherein a hard-fusion ED was used. The authors of [5] introduced a spectrum strategy using hypothesis testing as well as maximum a posteriori to differentiate the geosynchronous Earth orbit (GEO) signal from the interfering non-geostationary orbit (NGEO) and noise. On this basis, ref. [6] utilized the GLRT and maximum a-posterior criterion to derive the decision threshold to detect spectrum misuse in the satellite–terrestrial spectrum-sharing system. However, the basic idea behind the algorithms is the extension of GLRT to some extent. Overall, all these cooperative spectrum-sensing methods only achieve a limited improvement for spectrum-sensing performance with a single satellite, especially for sensing targets with a low signal to noise ratio (SNR). With such an insignificant improvement, it is difficult to meet the requirements of some targets with high sensitivity requirements. Moreover, these algorithms generally migrate the cooperative sensing algorithm from a terrestrial communication system directly to a satellite, without considering the particularity of the satellite system.

In contrast with terrestrial communication, satellites can cooperate with each other through exchanging their sensing data via inter-satellite links (ISL), which have a wide bandwidth and can complete almost error-free transmission in a vacuum environment. This unique characteristic provides new possibilities for satellite cooperation, so a distributed satellite cluster (DSC) at a short distance from each other is proposed. In general, a DSC is made up of several satellites in the same or adjacent orbit [7]. Compared with conventional satellite constellations, DSC has the advantages of low cost, high reliability and being a reconfigurable system. To the present, it has been widely used in three-dimensional imaging, meteorology, navigation and other fields [8,9]. While, in the field of communications, virtual beamforming based on DSC was proposed in [10], which improves the system capacity through the cooperation of a multi-satellite. With a reasonable attitude-control mechanism and powerful laser link, a DSC can exploit the spatial degree of freedom to improve the receiving gain of the target signal while degrading that of the interference signal [11,12]. Inspired by this, we use a DSC for spectrum sensing, in which the sensing signals are directly in-phase stacked, so as to enhance the sensing ability of weak signals by improving the received SNR.

Although a DSC is very attractive for sensing low SNR signals, the signal-level soft fusion requires time, frequency, and phase synchronization among satellites [13]. However, since the distance from a DSC to the ground is much greater than the distance between satellites, the time delay and Doppler frequency shift of signals received by satellites are approximately the same in a certain degree. In addition, some existing research has focused on time–frequency synchronization between satellites, and the proposed methods can meet the requirements of spectrum sensing [14,15,16].

In practice, the location information of the target is difficult to obtain in spectrum sensing. Coupled with the high dynamics and perturbation of the satellite, the phase offset from each concomitant to the master satellite cannot be known in advance. However, the beamforming gain is achieved depending on the phase alignment among satellites in the DSC. Thus, the key problem is to compensate the phase deviations. To some extent, the solution to this problem can be identified from the distributed and collaborative beamforming technology widely used in wireless sensor networks [17,18]. A one-bit feedback mechanism was proposed, wherein each transmitter adjusted its phase offset independently and iteratively to achieve carrier phase alignment at the distant receiver [19,20]. On this basis, a beamforming algorithm with a directional disturbance factor was presented in [21], which can track the direction of signal strength increases and improve convergence speed. However, since only the positive feedback information was utilized, the convergence speed and performance stability of these algorithms were insufficient. To overcome these limitations, a hybrid algorithm using successive negative-feedback information was studied in [22]. However, that algorithm only considers reducing the step size to improve the convergence speed, but, when the surrounding environment changes suddenly, the performance will deteriorate, resulting in poor applicability in high dynamic scenarios.

In this paper, we present a cooperative spectrum-sensing scheme based on the DSC system (SS_DSC) to enhance the sensing ability of weak signals with low SNR. A fast convergence algorithm that can select an appropriate perturbation size is proposed so as to improve the strength of the sensing target at the satellite receiver. In particular, the contributions of this article can be summarized as follows.

1.We conceive a general framework for cooperative spectrum sensing exploiting the DSC system. Based on this framework, a two-stage phase alignment method is conceived to improve the strength of the sensing signal. To be specific, in the first stage, the phase compensation is to eliminate the phase dynamics caused by satellite movement; the other is to eliminate the random phase deviations caused by satellite perturbation and non-ideal synchronization between satellites.2.We propose a two-bit feedback algorithm to eliminate the random phase in a second stage. Such an algorithm can achieve a faster convergence speed and a better stability, relying on adjusting the disturbance step according to the dynamic environment.3.We present a closed-form expression for the probability density function (PDF) of the sum of independent and identically distributed (i.i.d.) shadowed-Rician (SR) random variables. Based on the new PDF results, we derive the closed-form expression of the correct detection probability ($P_{d}$) for the considered DSC where the satellite link undergoes SR fading.

The organization of this paper is as follows. Section 2 describes the system model, based on which we analyze how to obtain the optimized beamforming weight vectors. Section 3 details the two-stage phase alignment frame. Section 4 derives the PDF expression for the sum of multiple i.i.d. SR RVs and the expression of $P_{d}$ for the considered SS_DSC system. The results and discussion are presented in Section 5 and this paper is concluded in Section 6.

## 2. System Model

We consider a SS_DSC system comprising of one master satellite and L−1 concomitant satellites for spectrum sensing, as shown in Figure 1. After receiving the task from the gateway station, each satellite of the DSC performs local sensing and sends the signal to the master satellite through the ISL for weighted fusion. However, in contrast with the traditional soft fusion methods, this method of fusion is to directly stack the sensing signals in phase, so as to greatly retain the effective information. Finally, the master satellite makes a judgment on the fused signal through energy detection, and transmits the result back to the Earth station to complete the whole sensing task.

In particular, our purpose is to obtain beamforming gain through signal-weighted fusion between the satellites to improve sensing performance. As mentioned earlier, the sensing target sends the signal *x* with $E(|x{|}^{2})=1$ to the DSC system through the fading channel. After weighted fusion, the output signal at master satellite can be written as
(1)$$s\left(t\right)=\sqrt{P_{t}}w^{H}\Phi hx\left(t\right)+w^{H}n_{0}\left(t\right)$$
where $P_{t}$ is the transmit power of the sensing target; $w^{H}=\left[e^{-j\phi_{1}\left(t\right)},e^{-j\phi_{2}\left(t\right)},\ldots ,e^{-j\phi_{L}\left(t\right)}\right]$ is the receive beamforming weight vector; $\Phi =diag\left[e^{j\Phi_{1}\left(t\right)},e^{j\Phi_{2}\left(t\right)},\ldots ,e^{j\Phi_{L}\left(t\right)}\right]$ is an L×L diagonal matrix where $\Phi_{i}\left(t\right)$ is the phase of the ith satellite; $n_{0}\left(t\right)$ denotes the additive white Gaussian noise (AWGN), which is an L×1 AWGN vector whose distribution is $\mathrm{CN}(0,\sigma_{n}^{2}I_{L\times 1})$; $h={\left[h_{1},h_{2},\ldots ,h_{L}\right]}^{T}$ is the real-valued channel vector between DSC system and sensing target modeled by the SR fading, which was proposed in [23], which illustrates similar agreements with the experimental data and has been used in most of the literature studies to model satellite channels of various radio frequency bands. The PDF of $h_{i}$ can be written as
(2)$$f_{h_{i}}\left(x\right)=2ax\exp\left(-\beta x\right){\;}_{1}F_{1}\left(m_{s};1;\delta x^{2}\right)$$
where $a={\left(2b_{s}m_{s}/2b_{s}m_{s}+\Omega_{s}\right)}^{m_{s}}/2b_{s}$, $\delta =\Omega_{s}{\left(2b_{s}\left(2b_{s}m_{s}+\Omega_{s}\right)\right)}^{-1}$, $\beta =1/2b_{s}$, with $\Omega_{s}$ represents the average power of the line-of-sight (LoS) component, $2b_{s}$ is the average power of the scatter component, and $m_{s}$ is the Nakagami parameter ranging from 0 to *∞*. Meanwhile, the function ${}_{1}F_{1}\left(a,b,z\right)$ is the confluent hypergeometric function, having expression as
(3)$${}_{1}F_{1}\left(a,b,z\right)=\sum_{k=0}^{\infty }\frac{{\left(a\right)}_{k}{\left(z\right)}^{k}}{{\left(b\right)}_{k}k!}$$
where ${\left(x\right)}_{k}=x\left(x+1\right)\cdots \left(x+k-1\right)$.

Thus, the output SNR of DSC can be written as
(4)$$\gamma =\frac{p_{t}}{\sigma_{n}^{2}}\frac{w^{H}\Phi hh^{H}\Phi^{H}w}{w^{H}w}=\frac{\bar{\gamma }}{L}|\sum_{i=1}^{L}\left|h_{i}\right|e^{j(\Phi_{i}\left(t\right)-\phi_{i}\left(t\right))}{|}^{2}$$
where $\bar{\gamma }=\frac{p_{t}}{\sigma_{n}^{2}}$ is the SNR at sensing target. Obviously, the SNR plays a key role in improving the spectrum-sensing performance. The higher the SNR, the higher the correct detection probability [24]. Mathematically, a constrained optimization problem for the design of the beamforming weight vectors can be formulated as
(5)$$\begin{array}{cc} \; & \max_{w}\gamma =\frac{p_{t}}{\sigma_{n}^{2}}\frac{w^{H}\Phi hh^{H}\Phi^{H}w}{w^{H}w}\\ & s.t.∥w∥=1 \end{array}$$

In general, the maximum ratio combining (MRC) scheme can obtain the optimal SNR; by applying the Cauchy–Schwarz inequality, it is not difficult to find that the optimal solution is given by
(6)$$w=\frac{\Phi h}{∥\Phi h∥}$$

However, the MRC reception requires full channel-state information (CSI) (i.e., both channel amplitude and phase) for all diversity branches, which is difficult to obtain for the SS_DSC system due to the lack of sensing-target location information. By comparison, equal gain combining (EGC) offers a somewhat reduced complexity alternative. In EGC reception, the sensing signals are weighted by phase only and the SNR at the master satellite is maximized when $w_{opt}^{H}=\left[e^{-j\Phi_{1}\left(t\right)},e^{-j\Phi_{2}\left(t\right)},\ldots ,e^{-j\Phi_{L}\left(t\right)}\right]$ is satisfied; then, we have
(7)$$\gamma_{opt}^{EGC}=\frac{\bar{\gamma }}{L}|\sum_{i=1}^{L}h_{i}{|}^{2}$$

Therefore, the optimization problem of maximizing SNR becomes finding the approximation of Φ, so as to achieve phase coherence at the master satellite. However, due to the random time-varying channel response, satellite high dynamics and non-ideal synchronization, the Φ can change in a random manner. In general, $\Phi_{i}\left(t\right)$ can be divided into two parts: one is the deviations between the master satellite and the concomitants, and the other is the deviations between DSC and ground sensing target. Thus, the phase deviations at the master satellite *i* at the *t* timeslot can be expressed as
(8)$$\Phi_{i}\left(t\right)=\underset{\mathrm{deviations}\;\mathrm{between}\;\mathrm{the}\;\mathrm{satellites}}{\underbrace{a_{i}\left(t\right)+b_{i}\left(t\right)+c_{i}\left(t\right)}}+\underset{\mathrm{deviations}\;\mathrm{between}\;\mathrm{DSC}\;\mathrm{and}\;\mathrm{sensing}\;\mathrm{target}}{\underbrace{\varphi_{i}\left(t\right)+\xi_{i}\left(t\right)}}$$
where $a_{i}\left(t\right)$ is caused by the change in the distance between the satellite and the ground, which leads to the initial phase difference between the master satellite and the concomitants; $b_{i}\left(t\right)$ is caused by the change in the satellite formation, and the change in distance between satellites leads to the rapid change of phase; and $c_{i}\left(t\right)$ is the random phase deviations caused by the perturbation of the DSC. Further, $\varphi_{i}\left(t\right)$ is the channel phase response from sensing target to the distant DSC, and $\xi_{i}\left(t\right)$ is the effect of non-ideal synchronization between them. Note that all values of the deviation variable are assumed to be uniformly distributed over [0,2π). Therefore, since Φ is caused by many reasons, we need to accommodate the remedy to the case for different reasons to obtain the approximate value of Φ, so as to obtain the $\gamma_{opt}^{EGC}$ at the master satellite.

## 3. The Proposed Two-Stage Phase Alignment Frame

### 3.1. Phase Compensation Analysis

In order to achieve carrier phase synchronization at the master satellite, each concomitant satellite in DSC adjusts its beamforming weighted phase $\phi_{i}\left(t\right)$ independently and iteratively. However, different methods can be used for different phase deviations. Herein, we conceive a two-stage phase alignment method for SS_DSC system so as to improve the strength of the sensing signal. To be specific, in the first stage, the phase compensation is to eliminate the phase dynamics caused by satellite motion, such as the aforementioned $a_{i}\left(t\right)$ and $b_{i}\left(t\right)$, while the second stage is to eliminate the random phase deviations caused by satellite disturbance and non-ideal synchronization, e.g., $c_{i}\left(t\right)$, $\varphi_{i}\left(t\right)$ and $\xi_{i}\left(t\right)$. Therefore, for $a_{i}\left(t\right)$, the phase compensation can be realized by broadcasting reference signal, i.e., the received sensing signal $s_{m}\left(t\right)$ from the master satellite to the concomitants. The phase compensation $\bar{a_{i}\left(t\right)}$ between the two signals is estimated in real time, which can be expressed as
(9)$$\bar{a_{i}\left(t\right)}=\arccos\left(R_{s_{m}s_{i}}\left(t\right)/\sqrt{R_{s_{m}s_{m}}\left(0\right)*R_{s_{i}s_{i}}\left(0\right)}\right)$$
where $R_{xy}\left(\cdot \right)$ denotes the correlation operation between *x* and *y*. Further, for $b_{i}\left(t\right)$, the phase compensation $\bar{b_{i}\left(t\right)}$ can be obtained by calculating the distance change $\Delta d_{im}\left(t\right)$ between the master satellite and its concomitants through the ephemeris, which can be expressed as
(10)$$\bar{b_{i}\left(t\right)}=\frac{2\pi }{\lambda }\Delta d_{im}\left(t\right)$$

However, in the second stage, it is difficult to calculate the random phase deviations directly through prior information. Thus, we need to design the adaptive weight component $\bar{u_{i}\left(t\right)}$ to eliminate randomness so as to achieve carrier phase synchronization at the master satellite. In this paper, we propose a random phase compensation algorithm based on two-bit feedback (RPC_TF) to obtain the value of $\bar{u_{i}\left(t\right)}$ without the knowledge of $c_{i}\left(t\right)$, $\varphi_{i}\left(t\right)$ and $\xi_{i}\left(t\right)$. In addition, it should be emphasized that there will still be some slight errors between satellite phases after $\bar{a_{i}\left(t\right)}$ and $\bar{b_{i}\left(t\right)}$ compensation, so $\bar{u_{i}\left(t\right)}$ is also their further compensation.

### 3.2. Two-Bit Feedback Algorithm

Note that the iterative speed and accuracy are the most important indicators for an iterative-type algorithm. Herein, the size of the disturbance is the key to controlling the convergence speed and stabilizing the performance of the phase compensation algorithm. Specifically, the perturbation size should be sufficiently large to improve the convergence speed during the initial stage. However, when close to the optimal point, the perturbation size should be reduced to achieve steady performance. Therefore, an excellent algorithm should have faster convergence speed and stability to adapt to the high dynamic characteristics of satellites. In view of this, an RPC_TF algorithm is proposed which introduces an additional direction correction factor $\tau_{i}\left(t\right)$ and a step control factor *q* to improve the convergence speed by changing the iteration step while ensuring the accuracy of the disturbance direction.

Without loss of generality, one slot in RPC_TF algorithm is set to start from the computation of the transmitting weight and end at the receiving of the two-bit feedback message from the receiver. During the convergence process, the master satellite sends two bits of information, $b_{0}$ and $b_{1}$, to the concomitant satellite in a time slot, and each concomitant synchronizes its phase based on these two bits.

Specifically, at time slot *t*, based on the weight fusion of the signals sent by all concomitant satellites, the master satellite compares the strength of the sensing signal $S_{S}\left(t\right)$ with its previously measured best strength $S_{S}^{best}\left(t-1\right)$. If $S_{S}\left(t\right)>S_{S}^{best}\left(t-1\right)$, the master satellite makes the first feedback bit $b_{0}=1$, updates the value $S_{S}^{best}\left(t\right)$ with the $S_{S}\left(t\right)$, and calculates the times of positive feedback, that is, the cumulative times of $b_{0}=1$. In addition, the second bit $b_{1}$ is set to 1 if the cumulative number of $b_{0}=1$ equals the preset threshold $\Gamma_{1}$ and to 0 otherwise. On the other hand, if $S_{S}\left(t\right)⩽S_{S}^{best}\left(t-1\right)$, the receiver sets $b_{0}=0$, maintains the previously measured best $S_{S}^{best}$, and also calculates the times of negative feedback, namely, the cumulative times of $b_{0}=0$. Similarly, the second bit $b_{1}$ is set to 1 if the cumulative number equals the preset threshold $\Gamma_{2}$ and to 0 otherwise.

Upon receiving the feedback from the master satellite, each concomitant satellite checks the two-bit information and makes a series of corresponding responses. The basic idea behind the proposed algorithm is that the concomitant satellites change the best phase compensation when the first bit $b_{0}=1$, and change the size of random perturbation when the second bit $b_{1}=1$, and vice versa. The proposed RPC_TF algorithm is summarized in Algorithm 1.
**Algorithm 1:**The RPC_TF algorithm.**A: At the master satellite****Input:** the sensing signal from the concomitant satellites $s_{i}\left(t\right)$, Initial value c=0, d=0**Two-bit information generation process:** The master satellite measures the strength of the sensing signal and updates the best strength and the bits.A1.  **If** $S_{S}\left(t\right)>S_{S}^{best}\left(t-1\right)$, **Then**
$S_{S}^{best}\left(t\right)=S_{S}\left(t\right)$, $b_{0}=1$, c=c+1;A2.      if $c=\Gamma_{1}$, then $b_{1}=1$, c=0A3.      else $b_{1}=0$A4.      end ifA5.  **Else** $S_{S}^{best}\left(t\right)=S_{S}^{best}\left(t-1\right)$, $b_{0}=0$, d=d+1;A6.      if $d=\Gamma_{2}$, then $b_{1}=1$, d=0A7.      else $b_{1}=0$A8.      end ifA9.  **End If****Output:** two-bit feedback information $b_{0}b_{1}$, the best strength $S_{S}^{best}\left(t\right)$.The master satellite sends $b_{0}b_{1}$ to the concomitant satellites.**B: At the concomitant satellites****Input:** direction correction factor $\tau_{i}\left(0\right)=0$, initial phase compensation $\bar{u_{i}\left(0\right)}=0$, step control factor q,q>1, random perturbation $\pm \delta_{0}$, two-bit feedback information $b_{0}b_{1}$.**Phase compensation process:** The transmitter obtains the feedback $b_{0}b_{1}$ from the receiver and updates its best transmission angleB1.  **If** $b_{0}b_{1}=10$, **then**        $\bar{u_{i,t}^{best}}=\bar{u_{i}}\left(t-1\right)+\delta_{i}\left(t-1\right)+\tau_{i}\left(t-1\right)$, $\tau_{i}\left(t\right)=\tau_{i}\left(t-1\right)$, $\tau_{i}\left(t\right)=\pm \tau_{0}\left(t-1\right)$;B2.  **Else if** $b_{0}b_{1}=11$, **then**        $\bar{u_{i,t}^{best}}=\bar{u_{i}}\left(t-1\right)+\delta_{i}\left(t-1\right)+\tau_{i}\left(t-1\right)$, $\tau_{i}\left(t\right)=\tau_{i}\left(t-1\right)$, $\tau_{i}\left(t\right)=\pm \tau_{0}\left(t-1\right)*q$;B3.  **Else if** $b_{0}b_{1}=01$, **then**        $\bar{u_{i,t}^{best}}=\bar{u_{i}}\left(t-1\right)$, $\tau_{i}\left(t\right)=-\delta_{i}\left(t-1\right)$, $\tau_{i}\left(t\right)=\pm \tau_{0}\left(t-1\right)/q$;B4.  **Else** $b_{0}b_{1}=00$, **then**        $\bar{u_{i,t}^{best}}=\bar{u_{i}}\left(t-1\right)$, $\tau_{i}\left(t\right)=-\delta_{i}\left(t-1\right)$, $\tau_{i}\left(t\right)=\pm \tau_{0}\left(t-1\right)$;B5.  **End If****Output:** the best phase compensation $\bar{u_{i,t}^{best}}$.Each concomitant satellites sends their signal towards the master satellite.**Repeat in** subsequent transmissions until convergence.

Consequently, the phase compensation $\bar{u_{i}\left(t\right)}$ is obtained to eliminate the randomness when RPC_TF algorithm is converged. The beamforming weighted phase $\phi_{i}\left(t\right)$ at each satellite can be achieve as $\phi_{i}\left(t\right)=\bar{a_{i}\left(t\right)}+\bar{b_{i}\left(t\right)}+\bar{u_{i}\left(t\right)}$. Therefore, we can obtain the beamforming gain with EGC reception, and the fusion SNR at the master satellite can be approximately equal to $\gamma_{opt}^{EGC}$. The proposed two-stage phase alignment frame diagram of SS_DSC system is summarized in Figure 2.

## 4. Sensing Performance Analysis

The final objective of this paper is to improve the sensing performance of weak signals. In this section, by using the method of probability theory, we derive the closed-form $P_{d}$ expression for the considered SS_DSC system.

There are two parameters associated with spectrum sensing: $P_{d}$ and the false alarm probability $P_{f}$, and energy detection is the most popular spectrum sensing scheme. Therefore, $P_{d}$ and $P_{f}$ for energy detection on the non-fading channel gain can be expressed as
(11)$$\begin{array}{cc} \; & P_{d}=Pr\left(Y>\lambda |H_{1}\right)=Q_{u}\left(\sqrt{2\gamma },\sqrt{\lambda }\right)\\ \; & P_{f}=Pr\left(Y>\lambda |H_{0}\right)=\frac{\Gamma (u,\frac{\lambda }{2})}{\Gamma (u)} \end{array}$$
where λ is detection threshold, *u* is the time–bandwidth product, $Q_{u}\left(a,b\right)$ represents the Marcum-Q function, and $\Gamma (u,\frac{\lambda }{2})$ refers to incomplete gamma function.

After the RPC_TF algorithm, the sensing signal of each satellite achieve phase alignment at the master satellite. At this time, the SNR γ is approximately equal to $\gamma_{opt}^{EGC}$. Thus, the PDF of γ can be shown as
(12)$$f_{\gamma }\left(\gamma \right)=\frac{1}{2}\sqrt{\frac{L}{\bar{\gamma }\gamma }}f_{Z}\left(\sqrt{\frac{L\gamma }{\bar{\gamma }}}\right)$$
where $Z=\sum_{i=1}^{L}h_{i}$, denotes the summation of *L* SR random variables. By substituting (Equation 12) in (Equation 11), $P_{d}$ can be expressed as
(13)$$P_{d}=\int_{\gamma }Q_{u}\left(\sqrt{2\gamma },\sqrt{\lambda }\right)f\left(\gamma \right)d\gamma =\frac{1}{2}\int_{\gamma }Q_{u}\left(\sqrt{2\gamma },\sqrt{\lambda }\right)\sqrt{\frac{L}{\bar{\gamma }\gamma }}f_{Z}\left(\sqrt{\frac{L\gamma }{\bar{\gamma }}}\right)d\gamma$$

From (Equation 13), we can see that the PDF of the combined signal amplitude *Z* is the key to solving the problem. Especially, the PDF of *Z* has been given as (Equation 2) when L=1, with the help of (Equation 2) and ([25], Equation (22)), the closed-form formula for $P_{d}$ under SR fading channel when L=1 can be obtained, yielding
(14)$$\begin{array}{cc} P_{d}^{1}=a\sum_{k=0}^{\infty }(\frac{{\left(m_{s}\right)}_{k}\delta^{k}}{{\left(1\right)}_{k}k!}( & \frac{\Gamma (k+1)}{\beta^{k+1}}-{\left(\frac{1}{\eta \bar{\gamma }}\right)}^{k+1}\exp\left(-\frac{\lambda }{2}\right)\\ \; & \times \sum_{j=u}^{\infty }\left[{\left(\frac{\lambda }{2}\right)}^{j}\frac{\Gamma (k+1)}{\Gamma (j+1)}{\;}_{1}F_{1}\left(k+1;j+1;\frac{\lambda u}{2\eta }\right)\right])) \end{array}$$
where $\eta =u+\frac{\beta }{\bar{\gamma }}$.

Next, we consider the summation of L(L⩾2) random variables. An accurate PDF expression of the amplitude of the combined signal is given by Theorem 1.

**Theorem** **1.**
*An accurate PDF expression of the sum of i.i.d. SR fading random variables $Z=\sum_{i=1}^{L},L⩾2$ is given by*

(15)
$$f_{L}\left(Z\right)=\Xi e^{-\beta Z^{2}}Z^{2\left(\sum_{i=1}^{L}k_{i}+\sum_{i=1}^{L-1}m_{i}\right)+2L-1}$$

*where*

(16)
Ξ=(2a)L∑k1=0∞(ms)k1δk1(1)k1k1!∏ρ=2L(∑kρ=0∞(ms)kρδkρ(1)kρkρ!∑t2ρ−3=02∑i=1ρ−1ki+∑i=1ρ−2mi+2ρ−32∑i=1ρ−1ki+∑i=1ρ−2mi+2ρ−3t2ρ−3∑t2ρ−2=02kρ+12kρ+1t2ρ−2(−1)t2ρ−3+(−1)t2ρ−22122∑i=1ρki+∑i=1ρ−2mi+2ρ−1∑mρ−1=0∞Γt2ρ−3+t2ρ−2+12Γt2ρ−3+t2ρ−2+32+mρ−1β2mρ−1)



The proof is relegated to Appendix A.

In Figure 3, the analytical and simulated PDFs of the sum of i.i.d. SR fading random variables, namely, *Z*, are plotted for different combinations of $\left(L,m_{s},b_{s},\Omega \right)$. It can be seen that the simulated PDF matches closely with the proposed analytical PDF we derived in (Equation 15).

Therefore, using (Equation 15) into (Equation 13), the closed-form expressions of $P_{d}^{L}$ of the i.i.d. SR faded L⩾2 branch EGC can be written as
(17)$$\begin{array}{cc} P_{d}^{L}= & \Xi \frac{1}{2}(\frac{\Gamma \left(\sum_{i=1}^{L}k_{i}+\sum_{i=1}^{L-1}m_{i}+L\right)}{\beta^{\sum_{i=1}^{L}k_{i}+\sum_{i=1}^{L-1}m_{i}+L}}-exp\left(-\frac{\lambda }{2}\right){\left(\frac{L}{\bar{\gamma }\eta }\right)}^{\sum_{i=1}^{L}k_{i}+\sum_{i=1}^{L-1}m_{i}+L}\\ \; & \sum_{j=u}^{\infty }{\left(\frac{\lambda }{2}\right)}^{j}\frac{\Gamma \left(\sum_{i=1}^{L}k_{i}+\sum_{i=1}^{L-1}m_{i}+L\right)}{\Gamma (j+1)}{\;}_{1}F_{1}\left(\sum_{i=1}^{L}k_{i}+\sum_{i=1}^{L-1}m_{i}+L;j+1;\frac{\lambda }{2\eta }\right)) \end{array}$$
where $\eta =u+\frac{L\beta }{\bar{\gamma }}$.

However, Equation (Equation 17) requires too much iterative time and calculative time, but when considering arbitrary integer-valued fading severity parameter [26,27], we can have
(18)$${}_{1}F_{1}\left(m_{s},1,\delta x\right)=e^{\delta x}\sum_{k=0}^{m_{s}-1}\frac{{\left(1-m_{s}\right)}_{k}{\left(-\delta x\right)}^{k}}{\Gamma {\left(1+k\right)}^{2}}$$
thus, the $P_{d}$ can be rewritten as: (19)$$\begin{array}{cc} P_{d}^{L}= & \Xi^{*}\frac{1}{2}(\frac{\Gamma \left(\sum_{i=1}^{L}k_{i}+\sum_{i=1}^{L-1}m_{i}+L\right)}{{\left(\beta -\delta \right)}^{\sum_{i=1}^{L}k_{i}+\sum_{i=1}^{L-1}m_{i}+L}}-exp\left(-\frac{\lambda }{2}\right){\left(\frac{L}{\bar{\gamma }\eta^{*}}\right)}^{\sum_{i=1}^{L}k_{i}+\sum_{i=1}^{L-1}m_{i}+L}\\ \; & \sum_{j=u}^{\infty }{\left(\frac{\lambda }{2}\right)}^{j}\frac{\Gamma \left(\sum_{i=1}^{L}k_{i}+\sum_{i=1}^{L-1}m_{i}+L\right)}{\Gamma (j+1)}{\;}_{1}F_{1}\left(\sum_{i=1}^{L}k_{i}+\sum_{i=1}^{L-1}m_{i}+L;j+1;\frac{\lambda }{2\eta^{*}}\right)) \end{array}$$
where $\eta =u+\frac{L(\beta -\delta )}{\bar{\gamma }}$,
(20)Ξ*=(2a)L∑k1=0ms−1(1−ms)k1(−δ)k1Γ(1+k1)2∏ρ=2L(∑kρ=0ms−1(1−ms)kρ(−δ)kρΓ(1+kρ)2∑t2ρ−3=02∑i=1ρ−1ki+∑i=1ρ−2mi+2ρ−32∑i=1ρ−1ki+∑i=1ρ−2mi+2ρ−3t2ρ−3∑t2ρ−2=02kρ+12kρ+1t2ρ−2(−1)t2ρ−3+(−1)t2ρ−22122∑i=1ρki+∑i=1ρ−2mi+2ρ−1∑mρ−1=0∞Γt2ρ−3+t2ρ−2+12Γt2ρ−3+t2ρ−2+32+mρ−1β−δ2mρ−1)

To the authors’ knowledge, the latter closed-form result is new and has not been reported elsewhere in the literature. Consequentially, the closed-form expression of $P_{d}$ for the proposed SS_DSC system is obtained. However, the $P_{f}$ is independent of SNR, which leads to the same $P_{f}^{L}$ for i=1,2,...,L as given by (Equation 11).

## 5. Simulation Results

In this section, we conduct computer simulations to evaluate the performance of the proposed algorithm and confirm the validity of our analytical results. The proposed two-stage phase alignment frame is simulated under Windows system. Satellite Tool Kit (STK) and MATLAB software are used as the main development tools, and the connect module inside STK is used as the interface between them. The hardware environment includes AMD Ryzen-7-5800 and GeForce GTX 1660 Ti. A DSC system is established in STK software to obtain the dynamic distance data between satellites and that between the satellites and sensing target in the process of movement. Then, the data is output to MATLAB software to convert the changing distance into the dynamic phase. The orbit parameters are shown in Table 1. Without loss of the generality, the satellite link is subject to SR average shadowing (AS) with $\left(m_{s},b_{s},\Omega \right)$ = (10.1, 0.126, 0.835) by default ([23], Table III), and the frequency band is 14 GHz (Ku).

Figure 4 illustrates the phase changes of the four concomitant satellites without phase control in 200 time slots. It can be found that the phases in the DSC system are randomly distributed. The reason for this is that the distance between each satellite and the sensing target, and the distance between satellites are constantly changing in the process of movement. Obviously, it is difficult to obtain beamforming gain if the singal-lever soft fusion is directly carried out at the master satellite. However, in Figure 5, the phases of the concomitant satellites gradually achieve synchronization from the initial random distribution with the help of the proposed RPC_TF algorithm, which proves the effectiveness of the algorithm.

Next, the average convergence speeds of the proposed RPC_TF algorithm, the directional perturbation feedback algorithm (DPA) in [21], and the modified hybrid algorithm (MHA) in [22] for the situation in which a sudden change occurs at timeslot *t* = 200 are depicted in Figure 6, where *L* = 4, $\bar{\gamma }$ = 0 dB, $\delta_{0}=\left(\frac{\pi }{5},\frac{\pi }{100}\right)$, respectively. Moreover, a maximum strength $S_{S}^{EGC}$ based on EGC is also given for comparison. It can be found that the three algorithms can converge to $\gamma_{EGC}$, and the error rate is less than 5%, which confirms the rationality of using EGC for theoretical analysis. To be specific, with a relatively large initial step size, $\delta_{0}=\frac{\pi }{5}$, the performance of the three algorithms is similar at the beginning, and RPC_TF is slightly faster. However, when a sudden change occurs, DPA can quickly recover and converge first, followed by RPC_TF, and MHA performs the worst. The reason for this is that DPA adopts a fixed step size and is not affected by the surrounding environment. However, the improvement in convergence speed is at the expense of performance: its convergence value is less than that of the other two algorithms. When it is a relatively small size $\delta_{0}=\frac{\pi }{100}$, the performances of RPC_TF and MHA are similar, and DPA is the worst in the initial stage.With the sudden change in environment, the proposed RPC_TF can recover rapidly, and only needs 95 time slots to reach 98% of the optimal value, while MHA requires less than 95% until 200 time slots; moreover, the convergence speed becomes slower and slower, which indicates that the proposed algorithm has faster convergence speed and stronger stability.

Next, we focus on the sensing performance of the proposed SS_DSC. Figure 7 illustrates the impact of satellite fading and the satellite number on the $P_{d}$ (equivalently probability of miss detection $P_{m}=1-P_{d}$ ) versus the false alarm probability, where number of sensing samples Num = 20, *L* = (3, 5), respectively. In addition, the satellite link follows the infrequent light shadowing (ILS) fading ($m_{s}$ = 20, $b_{s}$ = 0.158, Ω = 1.29), average shadowing (AS) fading ($m_{s}$ = 10, $b_{s}$= 0.126, Ω = 0.835), or frequent heavy shadowing (FHS) fading ($m_{s}$ = 1, $b_{s}$ = 0.063, Ω = 0.0007). As we see, an excellent agreement between the Monte-Carlo simulation and analytical results is achieved, which validates the derived closed-form expressions of the $P_{d}$. It is also clear that the sensing performance of *L* = 5 is better than that of *L* = 3 under equal conditions, demonstrating the benefits of employing more satellites in SS_DSC. Moreover, the $P_{m}$ for both the *L* = 5 and *L* = 3 clusters of ILS fading are lower than those of AS fading and FHS fading. This is because when the fading is serious, the average γ of satellites is lower; thus, the $P_{m}$ is higher.

In the case of the number of sampling points Num = 40, $P_{f}$ = 0.1, Figure 8 depicts $P_{d}$ for the proposed SS_DSC, RLRT, GLRT, ERD and OR algorithms versus SNR. It can be observed that the proposed SS_DSC is superior to other algorithms. Even when *L* drops to 3, it still performs best in several methods in the region of γ< –5, which shows the great potential of our proposed algorithm in the region of low SNR.

Figure 9 shows the receiver-operating-characteristic (ROC) curves of the five different algorithms where *L* = 5, the SNR γ = –10 dB and Num = 20. It is observed that the proposed SS_DSC outperforms other estimators, where the $P_{d}$ is 16.5% higher than that of RLRT at $P_{f}$ = 0.1. Even when the number of cooperative satellites *L* is reduced to 3, it is better than the traditional ERD, OR and GLRT methods, and the performance is improved by 36%, 41% and 74% at $P_{f}$ = 0.1, respectively. The reason for this is that the SS_DSC performs in-phase superposition after beamforming and retains the effective information of each sensing satellite as much as possible. However, the SS_DSC algorithm requires some time for phase alignment before spectrum sensing, while others do not. Therefore, it is necessary to flexibly select different sensing methods according to the needs of the scene. SS_DSC is suitable for the accurate sensing of some specific important targets, although it requires a higher implementation complexity.

However, the SS_ DSC has requirements for time synchronization between satellites, and Figure 10 shows the impact of time synchronization accuracy on cooperative SNR. It can be found that the SNR of the DSC system decreases gradually with the decrease in time synchronization accuracy. When the accuracy is at the level of 10^−14^, the impact of synchronization error on satellite cooperation is very small, but at the level of 10^−13^, the performance is only 66.3% when *L* = 2, 70.6% when *L* = 4, and 79.5% when *L* = 8, respectively. The reason is that the time synchronization error will change the initial phase of each satellite, and the weighted value obtained according to the perfect synchronization can not completely eliminate the phase difference, thus reducing the cooperation performance. In addition, the performance of DSC improves with the increase in the number of satellites. This is because the probability of weakening or even offsetting the time synchronization error increases in the process of signal fusion, which improves the robustness of the system to time synchronization. In short, time synchronization is very important to the system performance. Fortunately, the time synchronization accuracy of the current cesium atomic clock can reach the level of 10^−15^, which can meet the needs of SS_DCS.

## 6. Conclusions

In this paper, we present a general framework for cooperative spectrum sensing exploiting the DSC system to enhance the sensing performance of weak signals. Based on this framework, a two-stage phase alignment method is conceived to eliminate the dynamics and randomness of phase deviations respectively, so as to improve the strength of the sensing signal. Especially in the second stage, a two-bit feedback algorithm suitable for a satellite high dynamic environment is proposed. We present a closed-form expression for the PDF of the sum of i.i.d. SR RVs. Based on the new PDF expression, we derived the analytical expressions of $P_{d}$ for the considered SS_DSC system. Simulation results were provided to confirm the validity of the analytical results, which shows that the proposed scheme can significantly improve the $P_{d}$ especially for sensing weak signals, whilst achieving a faster convergence.

## Figures and Tables

**Figure 1 sensors-22-03983-f001:**
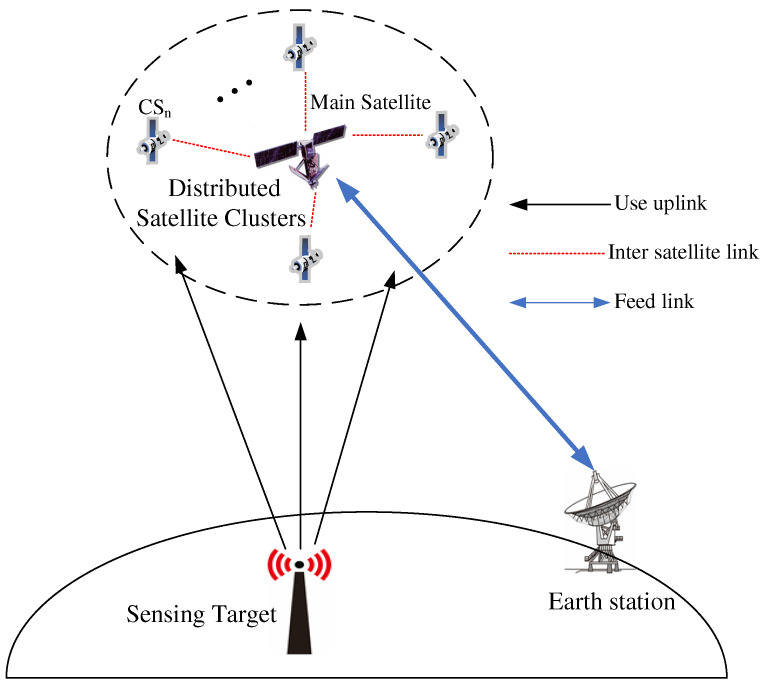
Proposed spectrum-sensing model comprising of a DSC system and a sensing target.

**Figure 2 sensors-22-03983-f002:**
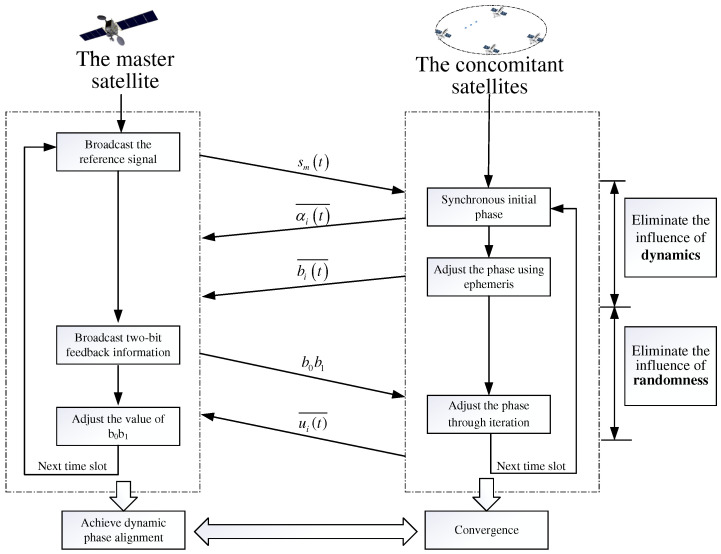
The two-stage phase alignment frame diagram for SS_DSC system.

**Figure 3 sensors-22-03983-f003:**
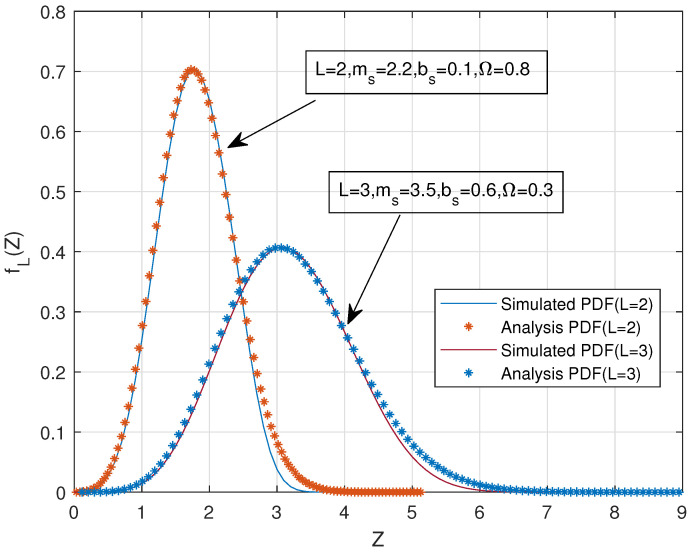
Analytical and simulated PDFs of the EGC scheme in SR fading links.

**Figure 4 sensors-22-03983-f004:**
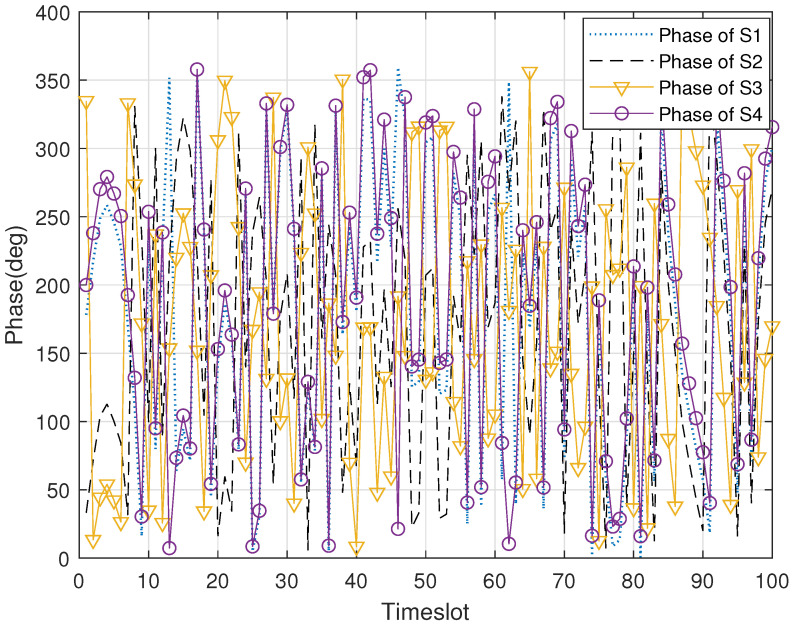
The signal phase of each concomitant satellite versus timeslot without phase control.

**Figure 5 sensors-22-03983-f005:**
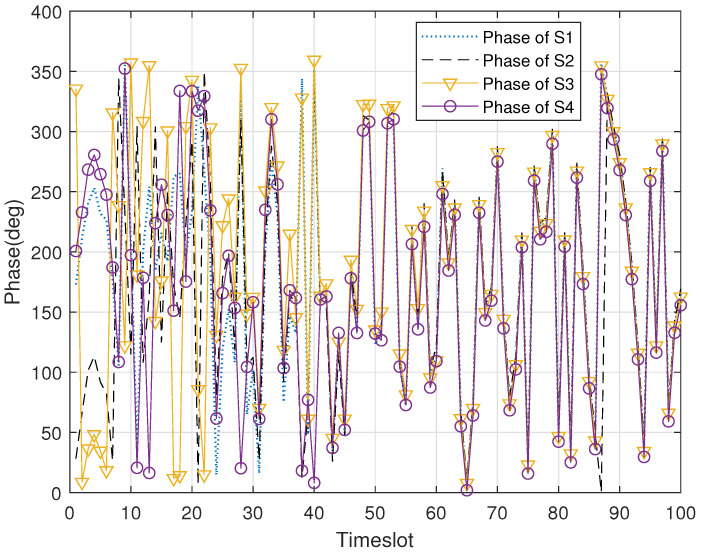
The signal phase of each concomitant satellite versus timeslot with RPC_TF algorithm.

**Figure 6 sensors-22-03983-f006:**
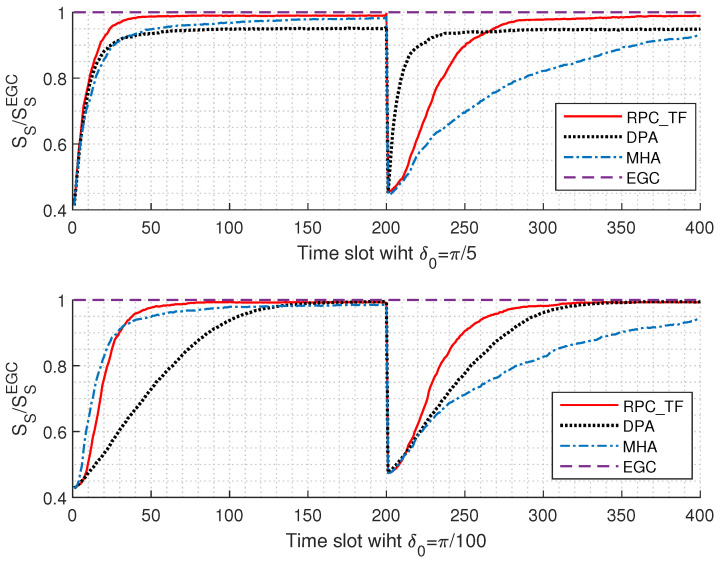
Comparison of the average convergence speeds of three algorithms for the situation in which a sudden change occurs at timeslot t=200, where *L* = 4, $\bar{\gamma }$ = 0 dB, $\delta_{0}=\left(\frac{\pi }{5},\frac{\pi }{100}\right)$, respectively.

**Figure 7 sensors-22-03983-f007:**
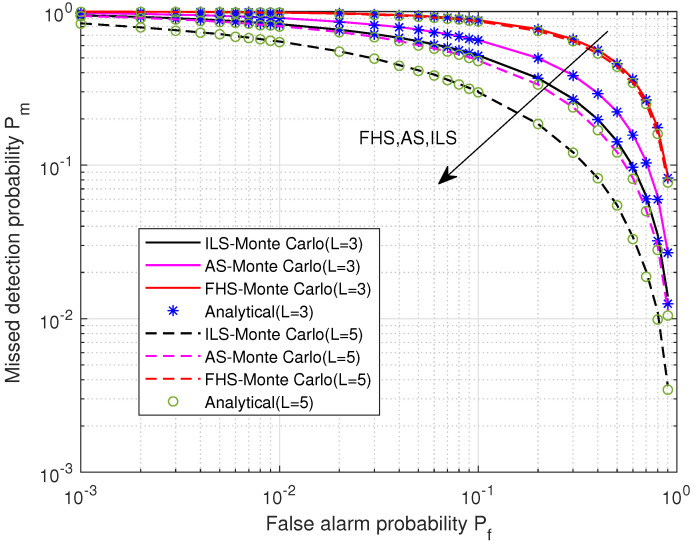
The probability of miss detection $P_{m}$ versus the false alarm probability $P_{f}$ for different shadowed-Rician fading values.

**Figure 8 sensors-22-03983-f008:**
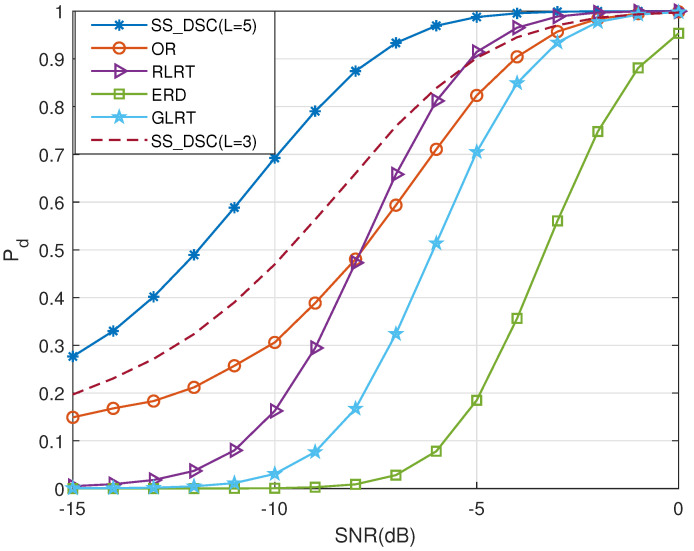
The probability of detection $P_{d}$ versus SNR for different algorithms when Num = 40 and $P_{f}$ = 0.1.

**Figure 9 sensors-22-03983-f009:**
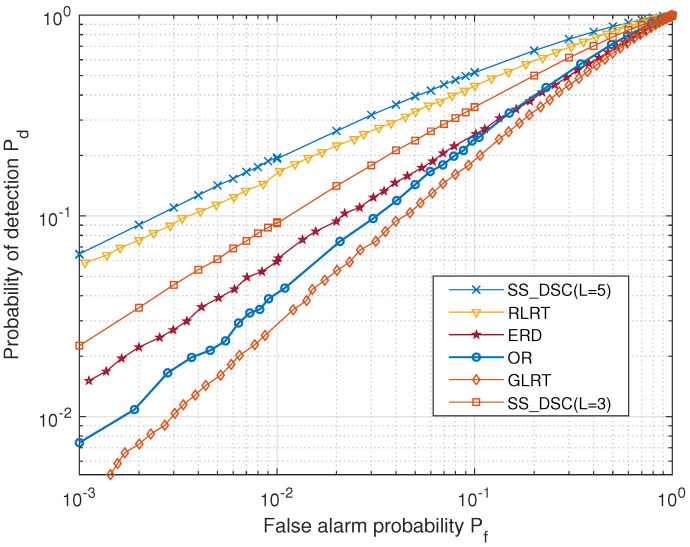
ROC curves of the five different algorithms where γ = –10 dB and Num = 20.

**Figure 10 sensors-22-03983-f010:**
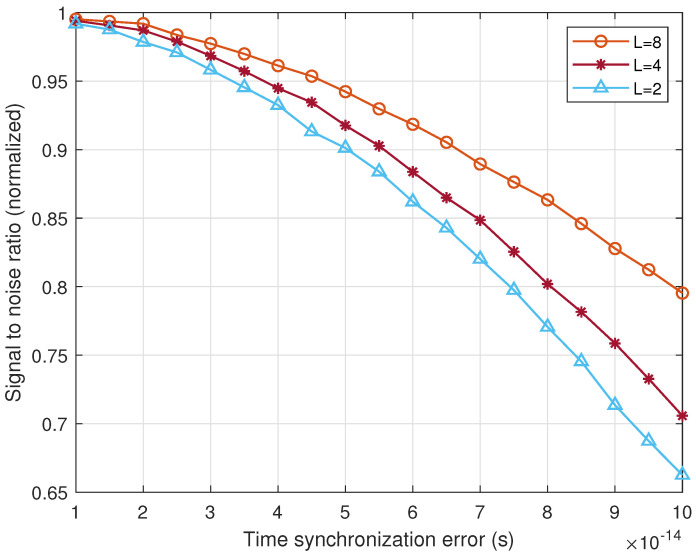
The impact of time synchronization accuracy on cooperative SNR where γ = 0 dB.

**Table 1 sensors-22-03983-t001:** Orbital parameters of distributed satellite clusters.

No.	Semimajor Axis	Eccentricity	Inclination Angle	Right Ascension of Ascending Node	True Anomaly
Master-Sat	7058.14 km	6.472 $\times \;10^{-16}$	68°	0°	11.917°
Concom-S1	7058.14 km	0.000356	68°	1.333 $\times \;10^{-14}$°	237.376°
Concom-S2	7058.14 km	0.000356	68°	1.117 $\times \;10^{-15}$°	147.434°
Concom-S3	7058.14 km	0.000356	68°	0°	57.246°
Concom-S4	7058.14 km	0.000356	68°	0°	327.148°

## Data Availability

Not applicable.

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
