# Peer review of "Distributed-Satellite-Clusters-Based Spectrum Sensing with Two-Stage Phase Alignment"

_sensors, 2022, doi:10.3390/s22113983_

Round 1
Reviewer 1 Report
The manuscript was well-prepared in theory and algorithm establishment. My only puzzle is that the establishment and validation of the whole algorithm is completely based on Satellite Tool Kit (STK) software simulation, how is its real applicability? As is well known, the difference between the software simulation results and the real situation may be quite large, so the practical applicability of the case analysis is crucial. Can it be test and shown to us? That will be more convincing for readers. If not, it is very essential to be clarified in the Discussion or Conclusion section.
Reviewer 2 Report
A distributed satellite clusters (DSC) system based spectrum sensing is used to optimize the ability for sensing weak signals. Comments:
- The literature review should be enriched by adding the relevant published papers. For example: WC Yeh, JS Lin, New parallel swarm algorithm for smart sensor systems redundancy allocation problems in the Internet of Things. The Journal of Supercomputing 2018, 74 (9), 4358-4384
- Abstract, “We investigate a distributed satellite clusters (DSC) system based spectrum sensing, …” a distributed satellite clusters (DSC) system is a method or a system studied?
- The abbreviations can only be mentioned after they defined, LEO not defined.
- The used computer environment not provided in simulation section.
- We cannot find the objective function.
6. How to prove the correctness of each Equation.
Round 2
Reviewer 1 Report
The revised manuscript has well answered my question.